# The Role of Nationality in Ultra-Endurance Sports: The Paradigm of Cross-Country Skiing and Long-Distance Running

**DOI:** 10.3390/ijerph17072543

**Published:** 2020-04-08

**Authors:** Beat Knechtle, Thomas Rosemann, Pantelis Theo Nikolaidis

**Affiliations:** 1Medbase St. Gallen Am Vadianplatz, 9001 St. Gallen, Switzerland; 2Institute of Primary Care, University of Zurich, 8091 Zurich, Switzerland; thomas.rosemann@usz.ch; 3Exercise Physiology Laboratory, 18450 Nikaia, Greece; pnikolaidis@uniwa.gr; 4School of Health and Caring Sciences, University of West Attica, 12243 Athens, Greece

**Keywords:** culture, endurance, genetics, nationality, performance, gender

## Abstract

Although the variation of performance by nationality in endurance sports such as marathon has been well studied, little information exists so far on the role of nationality on performance in ultra-endurance sports. The aim of the present study was to review the role of nationality on cross-country skiing and ultra-endurance running. Scopus and PubMed were searched using the syntax “nationality AND (ultra-endurance OR ultra-marathon OR cross-country skiing) in 1/4/2020. This search identified 17 articles, whose references were further examined for relevant literature. It was observed that Russian athletes dominated ultra-endurance running and cross-country skiing races. It was shown that these races were in other countries, where it was assumed that only the best Russians competed. Potential explanations could be misuse of performance enhancing substances, historical, climate-geographical and psychophysiological (e.g., combination of genetic and social factors). In summary, recent studies found a dominance of Russian athletes in specific races (i.e., ‘Comrades Marathon’, ‘Vasaloppet’, and ‘Engadin Ski Marathon’) and disciplines (i.e., ultra-marathon running, cross-country skiing) over a period of several decades. Future studies are need to investigate other events and other sport disciplines to confirm this Russian dominance.

## 1. Introduction

Recent studies have shown that Russian athletes dominate specific sports disciplines such as cross-country skiing (e.g., Engadin Ski Marathon, Vasaloppet) [1,2] and long-distance running (e.g., Comrades Marathon, 100-km ultra-marathons) [3,4]. These findings appeared to qualify for further investigation since in all data analyses always all finishers and—when possible—all years of all editions of these events were considered. In this review, we discussed these findings in detail and tried to provide potential explanations why Russian athletes dominate these specific disciplines or events. Scopus and PubMed were searched using the syntax “nationality AND (ultra-endurance OR ultra-marathon OR cross-country skiing) in 1/4/2020. This search identified 17 articles, whose references were further examined for relevant literature.

## 2. The Aspect of Nationality and Running Performance

It has been well-known since years that running races were dominated by athletes originating from specific regions of the world. Best running performances are dominated by a few groups including runners with West African ancestry for the sprint distances and East African runners for the long running distances up to the marathon [5]. Regarding long distance running, it was well known that East African runners from Kenya [6,7,8,9,10] and Ethiopia [11,12] dominate running races of different distances up to the marathon distance. In sprinting, runners from Jamaica were the best [13].

Potential explanations for the Kenyan dominance in running were both environmental and social factors. Most national and international Kenyan athletes came from the Rift Valley province, belonged to the Kalenjin ethnic group and Nandi sub-tribe, and spoke languages of Nilotic origin. A higher proportion of all athletes ran to school each day [9]. In Ethiopian runners, both environmental and genetic factors had an influence on athletic success. Elite endurance athletes were of a distinct environmental background in terms of geographical distribution and ethnicity. Ethiopian marathoners were mainly from the regions of Arsi and Shewa. The language distribution of marathoners differed from all groups with a predominance of languages of Cushitic origin. A higher proportion of marathoners ran to school and covered greater distances [12].

However, also for sprinters from Jamaica, success in sprinting might be related to both environmental and social factors. Sprint athletes predominantly originated from Surrey County, whilst middle distance athletes originated mainly from the Middlesex County. The language distribution of all groups showed a predominance of English. A higher proportion of both middle distance and jump and throw athletes walked to school and ran greater distances to school [13].

East African nationality seemed also to be important in pacing during a large city marathon. When data of 117,595 women and 180,487 men competing in the ‘New York City Marathon’ between 2006 and 2016 were analysed, Ethiopians and Kenyans were the fastest and runners from these countries showed more even pacing strategies than runners from other nationalities. Furthermore, the largest prevalence of an end spurt was in runners from the United States of America whereas the smallest prevalence was found in Ethiopians and Kenyans [14]. Overall, in running, colored athletes from Jamaica thus dominate sprint distances and from both Kenya and Ethiopia long-distance running (up to marathons).

## 3. Dominance in Cross-Country Skiing

Recent studies have reported that Russian athletes were the fastest in two cross-country skiing races, the ‘Engadin Ski Marathon’ held in Switzerland [2] and the ‘Vasaloppet’ held in Sweden [15] (Table 1). Like the fastest marathoners in the world (i.e., Ethiopians and Kenyans), the fastest cross-country skiers also showed the most even pacing. In the ‘Vasaloppet’, female and male Russians showed a more even pacing compared to their slower counterparts [15]. The dominance of Russians in the ‘Vasaloppet’ was confirmed in a recent analysis of this race from 1922 to 2017 [1].

## 4. Dominance in Long-Distance Running

In addition to cross-country skiing, it has been observed that Russian athletes were the fastest in long-distance running races such as the ‘Comrades Marathon’ held in South Africa [3] and in 100-km ultra-marathon running races [16] (Table 2). Ultra-marathoners from Russia were among the fastest in the world. In 100-km ultra-marathon running, the Russians, apart from the Japanese, were the most numerous athletes in the International Association of Athletics Federations (IAAF) since the IAAF has maintained a database of all the best results achieved in athletics worldwide [17]. An earlier investigation on 112,283 athletes (15,204 women and 97,079 men) from 102 countries who completed at least one 100-km ultra-marathon worldwide between 1998 and 2011 showed that the fastest 100-km ultra-marathoners were from Japan in both women and men [18]. A new investigation with data from 150,710 athletes who finished a 100-km ultra-marathon between 1959 and 2016 showed that most of the finishes were achieved by runners from Japan, Germany, Switzerland, France, Italy and United States of America but runners from Russia and Hungary were the fastest [16].

## 5. Potential Explanations for the Russian Dominance

A first potential explanation of Russian dominance could be that only the best Russian athletes competed in these races typically held very far away from Russia. In the analysis of the 100-km ultra-marathons held between 1959 and 2016, it was shown that runners from Russian achieved ~37% of the finishes outside of Russia but runners from Japan reached less than 2%. It was assumed that most probably only the fastest Russian runners competed outside of Russia in the fastest races such as World Championships or on the fastest courses worldwide. In contrast, Japanese runners competed preferably in races held in Japan where the courses were most probably not that fast [16].

This assumption might also explain why Russian athletes were the fastest runners in the ‘Comrades Marathon’ (South Africa) [3] and the fastest cross-country skiers in both the ‘Engadin Ski Marathon’ (Switzerland) [2] and the ‘Vasaloppet’ (Sweden) [15] although in all these races local athletes represented the highest numbers in participants. Also, in running races of 100 miles, a dominance of a specific nation has been reported. When all 100 miles ultra-marathons held between 1998 and 2011 were analyzed, most runners originated from the United States of America and the top ten US-American athletes achieved the fastest race times ever in women and men [19].

### 5.1. Origin of Athletes and Race Site in Triathlon

A similar finding that the fastest athletes in a specific sports discipline competed outside of their country of origin was reported for Ironman triathletes. When 302,535 athletes (61,087 women and 241,448 men) competing between 2002 and 2015 in 253 different Ironman triathlon races held worldwide were analysed, Germans were the fastest in both women and men [20]. The authors argued that German triathletes had a very large support in Germany with outstanding races such as ‘Challenge Roth’ or ‘Ironman Frankfurt’. Another potential explanation was that prior German winners of ‘Ironman Hawaii’ would share their experience with and support younger German athletes. For example, Patrick Lange, the winner of ‘Ironman Hawaii’ in 2018, was supported by Faris Al-Sultan, the winner of ‘Ironman Hawaii’ in 2005 [21].

However, there was a difference when the Ironman World Championship, the ‘Ironman Hawaii’ held in Hawaii (USA) was considered alone. When associations between nationalities and race times of 39,706 finishers originating from 124 countries competing in the ‘Ironman Hawaii’ from 1985 to 2012 were analyzed, most of the finishers originated from the United States of America followed by athletes from Germany and the fastest race times were achieved by US-American women and men [22]. In addition, US-Americans were also the fastest in the qualifiers for ‘Ironman Hawaii’ [23].

Although the Ironman triathlon was a US-American invention, a dominance of European athletes has been reported for triathletes competing in longer triathlon distances than the Ironman distance. In 1854 triathletes competing from 1985 to 2010 in 92 Double Iron ultra-triathlons (7.6 km swimming, 360 km cycling and 84.4 km running) held worldwide, the majority of the winners came from Europe with 72 victories, followed by North America with 17 victories. Race times of European athletes were faster than race times of American triathletes and Europeans improved their races over years compared to Americans [24]. An analysis of ultra-triathlons from Double to Deca Iron ultra-triathlon showed that the frequency of North Americans competing in Europe was very low (<5%), whereas Europeans often competed in North America (about 25%). Switzerland, France, and Germany were the fastest nations in ultra-triathlons [25].

### 5.2. Origin of Athletes and Race Site in Open-Water Swimming

Also in long-distance swimming, athletes of dominating nations such the United States of America and Australia in pool-swimming [26] were among the fastest in other events such as the ‘English Channel Swim’ [27,28]. In open-water swimming events such as the ‘Triple Crown of Open Water Swimming’, athletes from Australia, the USA and from the UK were among the fastest [29]. When swimming performances of successful swimmers in the ‘Triple Crown of Open Water Swimming’ (i.e., 535 athletes in ‘Catalina Channel Swim’, 1,606 athletes in ‘English Channel Swim’ and 774 athletes in ‘Manhattan Island Marathon Swim’) from 1875 to 2017 were analyzed, Australians were faster than US-Americans and US-Americans were faster than British and Canadians when considering all swimmers. When considering the annual five fastest, Australians were not faster than US-Americans but US-Americans were faster than British [29]. In the ‘English Channel Swim’, the most representative nations were United Kingdom, United States of America, Australia and Ireland. The fastest swim times were achieved by athletes from United States of America, Australia and the United Kingdom [27,30]. In other swimming races such as the long-distance open-water races from Fédération Internationale de Natation (FINA), no dominance of a particular nationality for all race distances was observed in races of 5 km, 10 km and 25 km, when all events held between 2000 and 2016 were considered [16].

### 5.3. Race Site and Local Athletes

In other events, preferably local athletes competed and also local athletes were among the most successful [16,31,32,33]. In the ‘Strait of Gibraltar’, the crossing between Europe (Spain) and Africa (Morocco) covering 14.3 km, Spanish swimmers were the most numerous and the fastest [16]. In ‘Norseman Xtreme Triathlon’ held between 2006 and 2014, most of the finishers originated from Norway and the fastest race times were achieved by Norwegian women and men [31]. In the 78-km ‘Swiss Alpine Marathon’, Swiss runners were the most numerous and the fastest runners originated from Switzerland for both women and men [32]. In ‘Powerman Duathlon World Championship’ held in Zofingen, Switzerland, from 2002 to 2011, European athletes from Switzerland and Germany dominated participation in the ‘Powerman World Championship’ where Swiss duathletes were the fastest [33]. In the ‘100 km Lauf Biel’, the oldest 100-km ultra-marathon in the world held since 1956 in Switzerland, athletes from Switzerland, Germany and France were among the most frequent participants throughout the history of the event [34].

### 5.4. The Aspect of Performance Enhancing Substances

Based upon news about doping in Russian athletes [35], the possibility of the use of performance enhancing drugs in these athletes should be considered [36,37]. The history of major incidents involving drug abuse by serious national players in sport suggested a 20-year cycle, with the former German Democratic Republic, China and now Russia employing similar strategies [38,39]. In the former German Democratic Republic, a state doping program was established [40]. From 1966 on, hundreds of physicians and scientists, including top-ranking professors, performed doping research and administered prescription drugs as well as unapproved experimental drug preparations in which mainly anabolic steroids were used [41]. Regarding economic structures, the German Democratic Republic was very similar to that in the Soviet Union (former Russia), with state ownership and centralized control [42]. We should also be aware that Russia, the former Soviet Union and the former German Democratic Republic had a very long tradition in research in exercise physiology [43,44].

During the Cold War between 1952 and 1988, the former Soviet Union dominated athletics [34]. One should be aware that the existence of a state sponsored sport doping program in Russia has recently been established. In November 2015, the World Anti-Doping Agency (WADA) implicated Russian athletes, coaches, trainers, doctors, laboratory directors, and sport officials in ‘a deeply rooted culture of cheating’ and ‘a systemic culture of doping’ [25]. The Russian Athletics Federation (RusAF) has been suspended by the IAAF since November 2015, after the World Anti-Doping Agency investigated allegations of widespread doping [45]. Russian athletes remain banned from international competitions after the IAAF upheld the country’s suspension into 2019 [46].

Regarding a potential use of performance enhancing drugs, we should consider the history of testing for prohibited substances. The fight against performance-enhancing substances began in the 1920s, long before there were testing methods for the prohibited substances [47]. The fight against doping in sports started as a result of the death of a Danish cyclist during the Rome Olympic Games in 1960 [48]. In 1967, the International Olympic Committee (IOC) established a Medical Commission responsible for developing a list of prohibited substances and methods. Drug tests were first introduced at the Olympic Winter Games in Grenoble and at the Olympic Summer Games in Mexico City in 1968 [49]. After the scandal at the 1998Tour de France, the fight against doping was intensified on national as well as international levels [50]. A major shift occurred in 1999, when the WADA was formed as an independent body to harmonize antidoping practices [51], and independent national antidoping organizations such as the Australian Sport Drug Agency, the Canadian Center for Ethics in Sport, and the United States Anti-Doping Agency (USADA) began to emerge [47,52,53]. WADA was established as a Swiss foundation with the support of intergovernmental bodies, governments, public authorities, and sports organizations [47,52].

We were not able to confirm the possibility for an abuse of performance enhancing drugs in ultra-marathoners and cross-country skiers competing in these races. Races such as the ‘Comrades Marathon’ [3], the ‘Engadin Ski Marathon’ [2] and the ‘Vasaloppet’ [15] have been held for decades and in some of the studies the time frames of the investigated periods started before official testing for prohibited substances started. Furthermore, testing for prohibited substances was primarily introduced for very large events such as Olympic Games or World Championships [49]. In addition, mainly high-level athletes were suspect of doping. Late in 1983, months before they announced a boycott of the Los Angeles Olympic Games, sports officials of the Soviet Union sent detailed instructions to the head of the nation’s track and field team [54].

Recent doping affairs clearly demonstrated that all sports were concerned, leading to a generalized suspicion concerning champions and their performance [55]. However, we must be aware that also recreational athletes may use performance enhancing drugs [56]. The prevalence of the use of prohibited drugs differed enormously and doping at an amateur level seemed to be less of a sport problem than a social problem [56]. There were also differences between sports disciplines. Risk of doping appeared to be highest in speed and power sports and lowest in motor skill- demanding sports [57].

### 5.5. Historical, Climate-Geographical and Psychophysiological Aspects

Other aspects that might explain the Russian dominance in ultra-endurance events included historical, climate-geographical and psychophysiological (combination of genetic and social factors). Through the elements of the historical heritage, the experience of the past, enshrined in the memory of living generations, as well as in historical documents could be analyzed to determine factors that probably influenced the formation and determination of the several traits specific to the Russian nation [58].

With regards to the climate-geographical aspect, the territory of Russia is vast, with very different climatic zones. The climate in this country—long winter, short summer, cold or hot steppe winds, an abundance of water in some areas, a lack of it in other areas—might have a great influence on the genotype of the people. In this context, a Russian might be considered to always strive for survival. Natural and climatic conditions over the centuries have shaped the performance, endurance and tolerance of the Russian people. The people were distinguished by the ability to concentrate physical forces at the right moment and perform super-efforts when it seemed that all human resources were exhausted. It should be noted that the labor during a year was characterized by a ‘broken’ rhythm of labor where most hard work was concentrated in the short summer period.

Eurasia, located at the junction of the continents of Asia and Europe, has been the scene of a large-scale ‘merger’ of several ethnic groups for thousands of years which also left an imprint on the formation of a national Russian character. Under the conditions of the North and Siberia, the life and work of people were largely associated with hunting and fishing, and working alone, which required strength, endurance and tolerance. The many-day lack of communication has ‘educated’ Russians to be closed, silent, and hard workers. From a socio-psychological point of view, patience and suffering were the most important values for Russians, along with consistent abstinence, self-restraint and constant self-sacrifice. Without these psychological traits, there would be no personality, no status and no respect by the others. These psychological traits supported the ‘eternal’ desire of a Russian to suffer, i.e., the desire for self-actualization, winning the inner freedom necessary to create good in the world and win the “freedom of the spirit”. In general, the world has been existed and moved only in terms of victims, patience and self-restraint. This was the cause of long-suffering, typical of Russian man, who could suffer a lot (especially material difficulties), if he knew why.

### 5.6. Genetic Background

Genetic differences between athletes from Russia and athletes from other countries might be considered [59,60,61]. All the listed above factors during the centuries probably brought to the determination of several gene polymorphism associated with endurance and capability to tolerate elongating fatigue. For example, the I allele of the angiotensin-converting enzyme (ACE) gene was associated with the endurance performance of the fastest 100 South African-born finishers in the 2000 or 2001 South African Ironman Triathlons [61]. A comparison of the frequency of distribution of genotypes and alleles in Russian athletes of various sport specializations and qualifications showed an association of the 964TT genotype for myogenic factor 6 (MYF6) with a predisposition to endurance exercise, the 577RR genotype and the 577R allele for actin alpha 3 (ACTN3)—with a predisposition to develop and show speed-power physical qualities and endurance [62,63]. We may also consider the possibility that the Russian dominance may be explained by the high probability of findings talents among its numerous populations, with respect to countries with less numerous populations.

## 6. Conclusions

In conclusion, recent studies have found a dominance of Russian athletes in specific competitions (i.e., ‘Comrades Marathon’, ‘Vasaloppet’, and ‘Engadin Ski Marathon’) and disciplines (i.e., ultra-marathon running, cross-country skiing) for several decades. Future studies may investigate other events and other sports disciplines to confirm a similar Russian dominance.

## Figures and Tables

**Table 1 ijerph-17-02543-t001:** Cross-country skiing races with dominance of Russian athletes.

Race	Time Frame	Number of Overall Subjects	Results
Vasaloppet [15]	2004–2017	183,919 finishers (19,465 women and 164,454 men)	In women, athletes from Russia (7:47:46 ± 1:41:38 h: min: s), Norway (7:51:52 ± 1:40:43 h: min: s), Austria (7:59:04±1:38:56 h: min: s), Estonia (8:14:03 ± 1:35:17 h: min: s) and Switzerland (8:15:14 ± 1:41:08 h: min: s) were the fastest. In men, athletes from Russia (6:33:03 ± 1:31:54 h: min: s) and Norway (6:36:33 ± 1: 37:01 h: min: s) were the fastest
Vasaloppet [1]	1922–2017	562,413 finishers	In women Russians, Norwegian, Austrians, and Estonians were the fastest. In men, Russians, Norwegians, and Austrians were the fastest
Engadin Ski Marathon [2]	1998–2016	197,125 finishers	Russians were the fastest

**Table 2 ijerph-17-02543-t002:** Ultra-marathons with dominance of Russian athletes.

Race/Events	Time Frame	Number of Overall Subjects	Results
Comrades Marathon [3]	1994–2017	235,467 finishers (40,211 women and 195,256 men)	In women and men, Russians were the fastest (12.55 ± 2.03 km/h and 12.24 ± 2.93 km/h, respectively). Also, Russians were the youngest (33.9 ± 4.6 and 36.3 ± 5.9 years, respectively).
100-km ultra-marathons worldwide [16]	1959–2016	150,710 finishers originating from 24 countries with a total of 307,871 finishes	Athletes from Russia achieved the fastest race times, when all athletes were considered by nationality.

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
