# Peer review of "The Role of Nationality in Ultra-Endurance Sports: The Paradigm of Cross-Country Skiing and Long-Distance Running"

_ijerph, 2020, doi:10.3390/ijerph17072543_

Round 1

Reviewer 1 Report

Thank you for your work. What an interesting piece you have offered. It seems the information is substantial and travels in a few directions. This makes your effort entertaining to consume and thought provoking. 

Abstract - 

Although informative, I do not believe the abstract reflect a concise summary of the manuscript. A reframing of the information to follow a more traditional presentation of the information is warranted. The current format is simply not concise enough and does not serve the purpose of an abstract. 

Section: (Section titles should be more consistent in their wording)

1: I believe the language can be elevated without being flamboyant. Terms such as 'very astonishing' are seemingly out of place. Perhaps 'interesting' or 'appear to qualify for further investigation'. 

Grammar needs cleaning throughout. 

2-5: Be mindful of the tense in which the information is offered. It seems to switch within the manuscript. 

Craft more consistent section heading titles. Russians or Russian athletes are the subject of the paper, perhaps the term does not need to be used seemingly at ever opportunity. 

Make sure all acronyms are introduced and then used. 

Would the PED section serve better as an independent section? 

6: This is a conclusion, not a summary. 

'Very recent' need not be used. 'Recent' will suffice. 

'partially several decades'. 'several decades or throughout several decades'

'need' changed to 'should' or 'may'

Overall, I believe there is interesting information provided in the manuscript. The authors have obviously worked diligently in its preparation and have displayed a deep understanding of the subject. Refining grammar, crafting a more consistent delivery and offering a perspective with consistent subject tense is required. 

Author Response

Thank you for your work. What an interesting piece you have offered. It seems the information is substantial and travels in a few directions. This makes your effort entertaining to consume and thought provoking. 

Abstract - 

Although informative, I do not believe the abstract reflect a concise summary of the manuscript. A reframing of the information to follow a more traditional presentation of the information is warranted. The current format is simply not concise enough and does not serve the purpose of an abstract. 

Answer: We agree with the expert reviewer and revised it (“Although the variation of performance by nationality in endurance sports such as marathon has been well studied, little information existed so far on the role of nationality on performance in ultra-endurance sports. The aim of the present study was to review the role of nationality on cross-country skiing and ultra-endurance running. Scopus and Pubmed were searched using the syntax “nationality AND (ultra-endurance OR ultra-marathon OR cross-country skiing) in 1/4/2020. This search identified 17 articles, whose references were further examined for relevant literature. It was observed that Russian athletes dominated ultra-endurance running and cross-country skiing races. It was shown that these races were in other countries, where it was assumed that only the best Russians competed. Potential explanations could be misuse of performance enhancing substances, historical, climate-geographical and psychophysiological (combination of genetic and social factors). In summary, recent studies found a dominance of Russian athletes in specific races (i.e. ‘Comrades Marathon’, ‘Vasaloppet’, and ‘Engadin Ski Marathon’) and disciplines (i.e. ultra-marathon running, cross-country skiing) over a period of several decades. Future studies need to investigate other events and other sports disciplines to confirm Russian dominance.”)

Section: (Section titles should be more consistent in their wording)

Answer:We changed the section titles where appropriate and added subtitles where appropriate.

1: I believe the language can be elevated without being flamboyant. Terms such as 'very astonishing' are seemingly out of place. Perhaps 'interesting' or 'appear to qualify for further investigation'. 

Answer:We changed to ‘These findings appear to qualify for further investigation since in all data analyses always all finishers and – when possible – all years of all editions of these events were considered’.

Grammar needs cleaning throughout. 

Answer:We checked the manuscript again for English grammar.

2-5: Be mindful of the tense in which the information is offered. It seems to switch within the manuscript. 

Answer:We changed now to past tense throughout the manuscript.

Craft more consistent section heading titles. Russians or Russian athletes are the subject of the paper, perhaps the term does not need to be used seemingly at every opportunity. 

Answer:We changed the section titles where appropriate and added subtitles where appropriate.

Make sure all acronyms are introduced and then used. 

Answer: We agree with the expert reviewer and revised the text for this aspect.

Would the PED section serve better as an independent section? 

Answer: We agree with the expert reviewer and revised it as independent section.

6: This is a conclusion, not a summary. 

Answer: We agree with the expert reviewer and revised it to “conclusion”.

'Very recent' need not be used. 'Recent' will suffice. 

Answer:We changed to ‘In summary, recent studies found a dominance of Russian athletes in specific races (i.e. ‘Comrades Marathon’, ‘Vasaloppet’, and ‘Engadin Ski Marathon’) and disciplines (i.e. ultra-marathon running, cross-country skiing) over a period of partially several decades’.

'partially several decades'. 'several decades or throughout several decades'

Answer:We changed to ‘In summary, recent studies found a dominance of Russian athletes in specific races (i.e. ‘Comrades Marathon’, ‘Vasaloppet’, and ‘Engadin Ski Marathon’) and disciplines (i.e. ultra-marathon running, cross-country skiing) throughout several decades

'need' changed to 'should' or 'may'

Answer:We changed to ‘Future studies may investigate other events and other sports disciplines to confirm Russian dominance

Overall, I believe there is interesting information provided in the manuscript. The authors have obviously worked diligently in its preparation and have displayed a deep understanding of the subject. Refining grammar, crafting a more consistent delivery and offering a perspective with consistent subject tense is required. 

Answer: We agree with the expert reviewer and revised the language of the text.

Reviewer 2 Report

Revision of the manuscript entitled:

The role of nationality in ultra-endurance: The paradigm of cross-country skiing and long-distance running

General comments

The study provides a discussion on the dominance of Russians in cross-country skiing and long-distance running. Apart from reporting data and findings of previous study, the Authors focus on a discussing potential factors that might contribute to explain such Russian dominance. The manuscript covers and interesting topic that is not only examined by a sport performance perspective, but also by historical, geographical and cultural perspectives. The manuscript is well written.

Did the Authors consider the possibility that the Russians dominance may be explained by the high probability of findings talents among its numerous population, with respect to countries with less numerous population?

I have also some specific comments that I hope will be useful to improve its scientific quality. However, I suggest to check the language of the text, as several typos are present.

Specific comments

Line 17. Insert a comma between “other sports” and “the role”

Line 38. Modify as “Regarding long distance running, it is well known that…”

Line 45. Modify as “…and genetic factors have…”

Line 45-46. Not clear. Please rephrase.

Line 49. I suggest to begin a new paragraph with “However…”.

Line 55. “…and 180,487” what? Men?

Line 59. I suggest to add sentences closing this subchapter of the review, thus synthesizing its content.

Line 65. “the most even pacing”? not “event”?

Line 75. Modify adding commas as “…the Russians are, apart from the Japanese, the most numerous…”

Line 78. Substitute “about” with “on”

Line 80-82. Please rephrase. This notion is already state in the previous sentence.

Line 84. “finishers” instead of “finishes”?

Line 90. Modify as: “A first potential explanation of Russian dominance could be that…”

Line 93. “…runners from Japan were less than 2%”?

Line 96-99. Sentence too long. Please rephrase.

Line 151. Caution. “should” instead of “must”.

Line 163. “…sport doping program…”

Author Response

Reviewer 2

Revision of the manuscript entitled:

The role of nationality in ultra-endurance: The paradigm of cross-country skiing and long-distance running

General comments

The study provides a discussion on the dominance of Russians in cross-country skiing and long-distance running. Apart from reporting data and findings of previous study, the Authors focus on a discussing potential factors that might contribute to explain such Russian dominance. The manuscript covers and interesting topic that is not only examined by a sport performance perspective, but also by historical, geographical and cultural perspectives. The manuscript is well written.

Answer:We thank the expert for his/her comment.

Did the Authors consider the possibility that the Russians dominance may be explained by the high probability of findings talents among its numerous populations, with respect to countries with less numerous population?

Answer:We added this aspect at the end of the discussion.

I have also some specific comments that I hope will be useful to improve its scientific quality. However, I suggest to check the language of the text, as several typos are present.

Answer:We checked the manuscript again for English grammar.

Specific comments

Line 17. Insert a comma between “other sports” and “the role”

Answer:We changed as suggested

Line 38. Modify as “Regarding long distance running, it is well known that…”

Answer:We changed as suggested

Line 45. Modify as “…and genetic factors have…”

Answer:We changed as suggested

Line 45-46. Not clear. Please rephrase.

Answer:We changed to ‘Elite endurance athletes are of a distinct environmental background in terms of geographical distribution and ethnicity’.

Line 49. I suggest to begin a new paragraph with “However…”.

Answer:We changed as suggested

Line 55. “…and 180,487” what? Men?

Answer:We added ‘men’

Line 59. I suggest to add sentences closing this subchapter of the review, thus synthesizing its content.

Answer:We added ‘Overall, in running, colored athletes from Jamaica were dominating sprint distances and from both Kenya and Ethiopia in long-distance running (marathon)’ to that section.

Line 65. “the most even pacing”? not “event”?

Answer:We changed as suggested

Line 75. Modify adding commas as “…the Russians are, apart from the Japanese, the most numerous…”

Answer:We changed to ‘In 100-km ultra-marathon running, the Russians, apart from the Japanese, were the most numerous athletes in the IAAF (International Association of Athletics Federations) since the IAAF has a database of all the best results achieved in athletics worldwide’.

Line 78. Substitute “about” with “on”

Answer:We changed as suggested.

Line 80-82. Please rephrase. This notion is already state in the previous sentence.

Answer:We deleted the sentence and changed to other to ‘An earlier investigation on 112,283 athletes (15,204 women and 97,079 men) from 102 countries who completed a 100-km ultra-marathon worldwide between 1998 and 2011 reported that the fastest 100-km ultra-marathoners were from Japan in both women and men’.

Line 84. “finishers” instead of “finishes”?

Answer:We have to write ‘finishes’ since the number of finishes races was considered, not the number of finishers.

Line 90. Modify as: “A first potential explanation of Russian dominance could be that…”

Answer:We changed to ‘A first potential explanation of Russian dominance could be that only the best Russian athletes compete in these races partially very far away from Russia’.

Line 93. “…runners from Japan were less than 2%”?

Answer:We changed to ‘In the analysis of the 100-km ultra-marathons held between 1959 and 2016, it was shown that runners from Russian achieved ~37% of the finishes outside of Russia but runners from Japan reached less than 2%’.

Line 96-99. Sentence too long. Please rephrase.

Answer:We changed to ‘This assumption might also explain why Russian athletes were the fastest runners in the ‘Comrades Marathon’ (South Africa) [3] and the fastest cross-country skiers in both the ‘Engadin Ski Marathon’ (Switzerland) [2] and the ‘Vasaloppet’ (Sweden) [15] although in all these races local athletes achieved the highest numbers in participants’.

Line 151. Caution. “should” instead of “must”.

Answer:We changed as suggested.

Line 163. “…sport doping program…”

Answer:We changed as suggested.

Round 2

Reviewer 1 Report

I applaud the versatility of the authors and their energy in making suggested revisions. The quality of the work has improved with an enhanced style of communication. 

Well done.